# DRFusion: Drift-Resilient Temporally Consistent Infrared–Visible Video Fusion

Xingyuan Li [1]  Haoyuan Xu [2]  Shulin Li [2]  Xiang Chen [1]  Zhiying Jiang [3]  Jinyuan Liu [2]

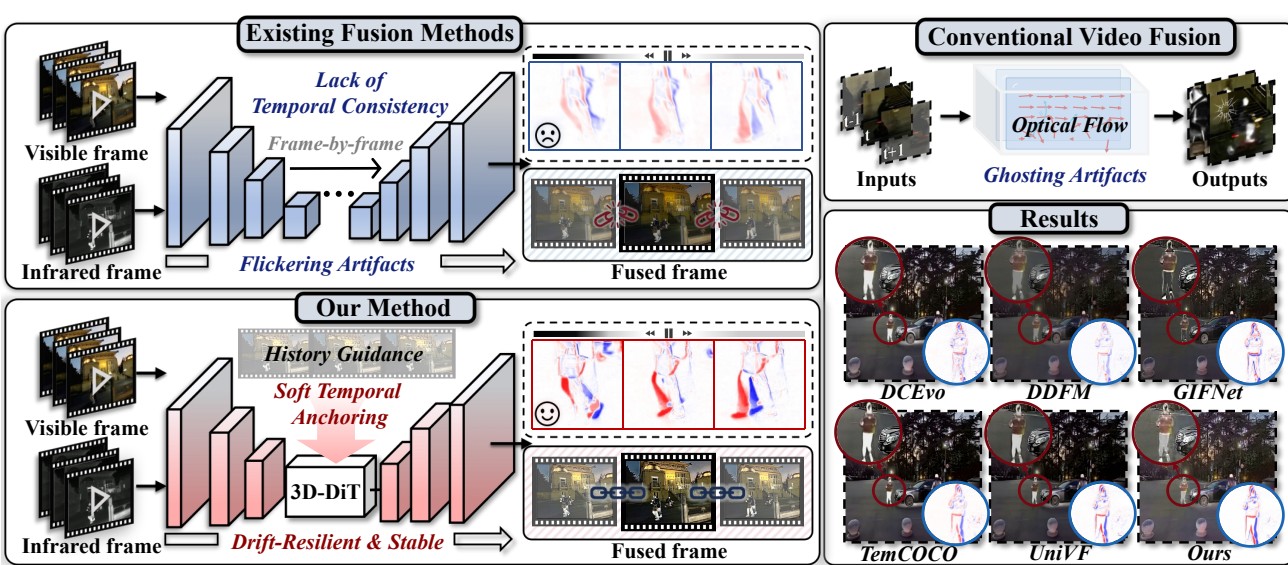

*Figure 1.* Frame-by-frame fusion methods are prone to temporal jitter, whereas Conventional optical flow-based methods often introduce "geometric rigidity". In contrast, our DRFusion leverages Stabilized History Guidance and Decoupled Structure-Motion Adaptation strategies to effectively overcome these limitations, achieving superior performance in both fusion quality and temporal stability.

## Abstract

Infrared and visible video fusion is essential for achieving comprehensive perception in dynamic scenes. However, maintaining temporal consistency remains a formidable challenge. Conventional methods relying on optical flow often suffer from geometric rigidity and ghosting artifacts. Moreover, standard diffusion-based fusion models typically operate in a frame-by-frame manner; when extended to autoregressive settings, they lack intrinsic temporal constraints and are prone to severe error accumulation and drifting, where minor artifacts amplify over time. To address these limitations, we propose a drift-resilient

video fusion method that reformulates the task as history-conditioned motion generation. We introduce Stabilized History Guidance and Soft Temporal Anchoring to reframe temporal consistency as spectral filtering, implicitly aggregating motion dynamics without rigid alignment. Furthermore, our Decoupled Structure-Motion Adaptation strategy bridges pre-trained priors and structural constraints via two-stage training and latent refinement. Extensive experiments demonstrate that our method achieves state-of-the-art performance in both fusion quality and temporal stability. Code is available at https://github.com/xhhaoyan/DRFusion

[1] College of Computer Science and Technology, Zhejiang University, Hangzhou, China  [2] School of Software Technology & DUT-RU International School of ISE, Dalian University of Technology, Dalian, China  [3] College of Information Science and Technology, Dalian Maritime University, Dalian, China . Correspondence to: Jinyuan Liu <atlantis918@hotmail.com>.

*Proceedings of the 43rd International Conference on Machine Learning*, Seoul, South Korea. PMLR 306, 2026. Copyright 2026 by the author(s).

## 1. Introduction

Image fusion has emerged as a cornerstone technique in the fields of computer vision and image processing, aiming to integrate complementary information from distinct sources into a single, information-rich representation (Zhao et al., 2024). In particular, infrared and visible image fusion has garnered significant attention (Li et al., 2025a; Liu et al.,

2025b). Visible sensors excel at capturing rich texture details under sufficient lighting conditions, whereas infrared sensors are sensitive to thermal radiation, effectively highlighting salient targets in low-light environments or adverse weather conditions. Consequently, fusing these two modalities demonstrates immense potential in applications such as military detection (Liu et al., 2022; Li et al., 2024b), security surveillance (Yi et al., 2024; Bai et al., 2025), and autonomous driving (Liu et al., 2023; Li et al., 2023b). In real-world scenarios, visual perception is inherently continuous and dynamic. Static fusion approaches that process isolated images fail to capture the temporal continuity required for practical deployment. Therefore, infrared and visible video fusion is crucial (Zhao et al., 2025); it not only preserves critical spatial information but also maintains temporal consistency, offering a more comprehensive representation of the scene.

Current research remains predominantly focused on static images (Li et al., 2026; Guan et al., 2026; Liu et al., 2026), witnessing an evolution in deep learning architectures from early CNN and GAN-based (Ma et al., 2020; Cao et al., 2024) methods to sophisticated Transformer models capable of capturing global dependencies (Ma et al., 2022; Yi et al., 2024). Furthermore, diffusion-based approaches leverage generative priors to effectively align distinct cross-modal distributions (Wang et al., 2025; Yue et al., 2023). However, adapting these architectures—optimized for static scenes—to dynamic video sequences presents a significant challenge. Existing methods often treat video sequences merely as collections of independent images, neglecting inter-frame correlations. Such frame-independent processing leads to unstable temporal fluctuations in brightness and texture, resulting in severe visual flickering artifacts that can disrupt downstream video analysis algorithms.

Second, although existing video fusion methods attempt to incorporate temporal information, early attempts relying on simple mechanisms like temporal averaging or basic RNNs fail to capture non-linear motion dynamics (Xie et al., 2024a; Gong et al., 2025). This typically results in over-smoothing and motion blur. To enhance temporal consistency, more recent approaches have introduced "explicit temporal constraints" via optical flow (Zhao et al., 2025; Li et al., 2024a; Zhao et al., 2021). However, this strategy introduces a fundamental limitation known as "geometric rigidity." Unlike soft attention mechanisms, rigid warping operations are highly sensitive to estimation errors, particularly in scenarios involving rapid motion or occlusion. Forcibly imposing such hard constraints on imprecise alignments inevitably results in structural distortions and ghosting artifacts.

Simultaneously, existing diffusion-based fusion methods (Zhao et al., 2023c) still encounter limitations in dynamic video scenarios. First, they lack intrinsic temporal modeling capabilities. Relying on 2D architectures (such as U-Net), these models typically process videos in a frame-by-frame manner, inherently ignoring inter-frame correlations, which results in severe temporal jitter and incoherence. Second, they face an "Adaptation Dilemma" when extending to video tasks. How to effectively incorporate native video generative priors into fusion tasks—fully utilizing temporal dynamics while aligning with structural constraints—remains a pressing research problem.

To address these challenges, we propose **DRFusion**, a **D**rift-**R**esilient 3D-DiT framework tailored for temporally consistent infrared-visible video fusion. Distinct from previous image-based diffusion approaches that lack intrinsic temporal modeling capabilities and process videos in a frame-by-frame manner, we reformulate video fusion as a **history-conditioned motion generation process** adapted from a pre-trained video foundation model. Our core insight is to resolve the "adaptation dilemma" via a **decoupled two-stage training strategy**. We freeze the robust motion priors of the 3D-DiT and employ a lightweight adapter to bridge the domain gap between the generative manifold and structural constraints. More critically, we treat temporal consistency as a spectral filtering problem. By introducing a "Soft Temporal Anchoring" mechanism, the model implicitly aggregates motion dynamics from a stabilized history. This design mathematically isolates robust low-frequency motion trends from high-frequency error accumulation, effectively breaking the feedback loop inherent in autoregressive generation to ensure long-term stability. Our main contributions are as follows:

- We propose DRFusion, a history-conditioned generative framework that reformulates video fusion as a probabilistic motion generation process. This paradigm effectively resolves the intrinsic conflict between temporal consistency and structural fidelity, eliminating the geometric distortions inherent in traditional optical flow-based methods.

- We introduce Stabilized History Guidance integrated with Soft Temporal Anchoring. By theoretically interpreting noise injection as a spectral filtering process, we isolate robust low-frequency motion trends from high-frequency autoregressive errors, ensuring drift-free long-term stability.

- We design a Decoupled Structure-Motion Adaptation strategy rooted in a two-stage training framework. By synergizing a learnable Condition Adapter with inference-time Latent Refinement, we bridge the gap between pre-trained motion priors and infrared structural constraints in a "coarse-to-fine" manner.

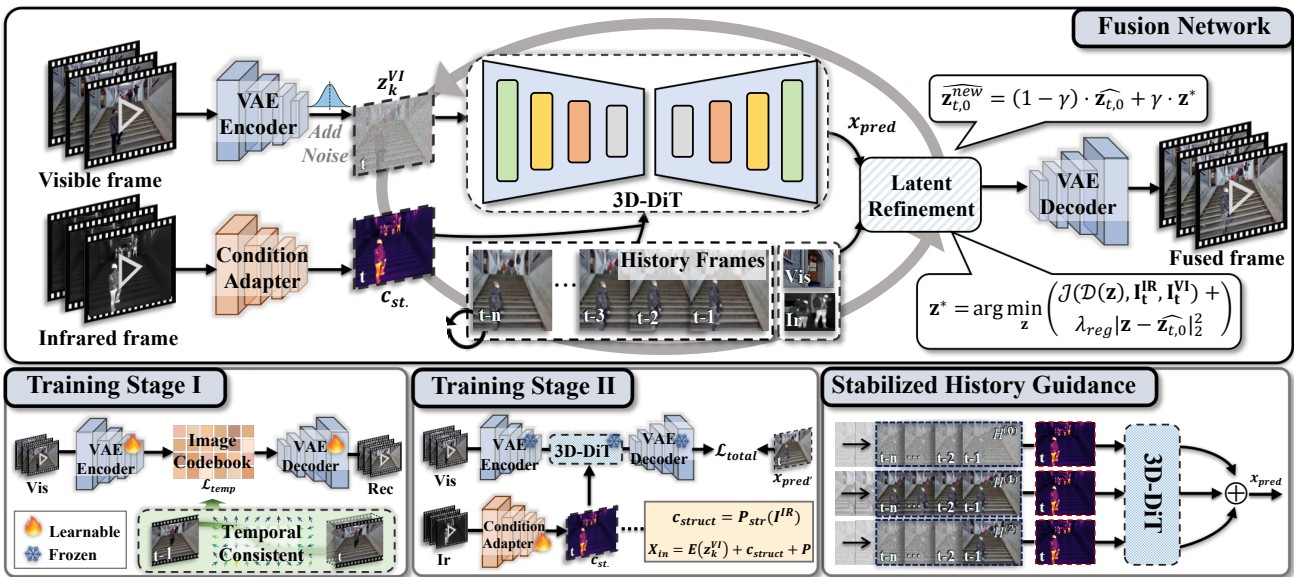

*Figure 2.* Overview of DRFusion.

## 2. Related Works

### 2.1. Infrared and visible image fusion

Deep learning has significantly advanced IVIF, moving beyond traditional signal processing to learn data-driven representations. Early autoencoders (Zhao et al., 2020; Li et al., 2021) and CNNs (Xu et al., 2022; Zhao et al., 2023a) improved feature extraction but struggled with global context. Transformer-based models (Tang et al., 2022) utilize self-attention to capture global interactions, while recent State Space Models like Mamba (Xie et al., 2024b) offer a linear-complexity alternative. Hybrid frameworks (Zhao et al., 2023b) further integrate these global mechanisms with local CNN features, balancing thermal structure preservation with textural detail.

Currently, the research paradigm is shifting toward generative modeling and semantic guidance. Diffusion models (Zhao et al., 2023c; Yue et al., 2023) utilize generative priors to align the distinct distributions of infrared and visible domains. Furthermore, incorporating natural language has emerged as a key direction. Methods like Text-IF (Yi et al., 2024) and PromptFusion (Liu et al., 2024a) introduce text prompts for interactive control, while CLIP-based approaches (Wang et al., 2024) align fused results with high-level semantic concepts, prioritizing semantic understanding over simple pixel reconstruction.

### 2.2. Diffusion-based video fusion

Frame-by-frame fusion often leads to flickering and inconsistency due to neglected temporal dependencies. . To address this, traditional video restoration approaches ( (Zhou et al., 2024)) exploit inter-frame features to maintain continuity. For video fusion specifically, frameworks such as (Gong et al., 2025; Zhao et al., 2025; Li et al., 2025b) integrate explicit temporal modeling or optical flow to align temporal features. However, these methods often suffer from high computational costs and complex processing steps. Diffusion models have recently shown strong generative performance, but their adaptation to video tasks remains non-trivial. Direct application of image-based diffusion often leads to temporal incoherence (Singer et al., 2022). While 3D architectures like (Ho et al., 2022) ensure consistency, they demand excessive computational resources. Consequently, efficient strategies such as one-shot tuning (Wu et al., 2023) or motion adapters (Guo et al., 2023) have emerged to adapt image models for video tasks. Despite these advancements, effectively leveraging such efficient generative priors for video fusion, which requires balancing high inference speed with signal fidelity, remains a significant research gap.

## 3. Methodology

We propose DRFusion, a Drift-Resilient 3D-DiT framework that reformulates video fusion as a history-conditioned probabilistic motion generation process. Given infrared $V_{IR} = \{I_t^{IR}\}$ and visible $V_{VI} = \{I_t^{VI}\}$ sequences, we operate within a unified spatiotemporal latent space to generate a fused sequence $V_F = \{F_t\}$ that preserves multimodal saliency. By learning the conditional distribution of the current frame given history, our method implicitly captures motion dynamics—ensuring temporal consistency without explicit geometric alignment. The overall pipeline is illustrated in Figure 2.

## 3.1. Decoupled Structure-Motion Adaptation

To synthesize fused video sequences via the iterative latent denoising process, we establish a Two-Stage Temporal-Conditional Training framework. This strategy decouples the optimization of the representation space from generative modeling. We first fine-tune a temporally constrained VAE to ensure a flicker-free latent foundation (Stage I), and subsequently employ a self-supervised strategy to train a lightweight adapter (Stage II), enabling the frozen 3D-DiT to achieve high-quality fusion by aligning the generative process with infrared structural priors.

### 3.1.1. STAGE I: TEMPORALLY-CONSTRAINED VAE

To construct a unified spatiotemporal latent space, we extend the standard image-based VQ-VAE with a temporal smoothness constraint. Here, optical flow is introduced solely as a soft regularization term to align the latent manifold. This design encourages the encoder to learn temporally continuous representations, enhancing the temporal consistency of the latent space. Specifically, we employ a pre-trained optical flow estimator (Ranjan & Black, 2017) to compute the motion field $w_{t-1 \to t}$ from the previous frame to the current one, and minimize the occlusion-aware latent warping error:

$$\mathcal{L}_{temp} = \sum_{t=2}^{T} \frac{\|M_t \odot (W(z_{t-1}, w_{t-1 \to t}) - z_t)\|_2^2}{\|M_t\|_1 + \epsilon}, \quad (1)$$

where $M_t$ denotes the occlusion mask, and $W(\cdot)$ represents the warping operation at the latent resolution. The VAE is optimized end-to-end via the following compound objective:

$$\mathcal{L}_{VAE} = \mathcal{L}_{rec} + \lambda_{adv}\mathcal{L}_{GAN} + \lambda_{vq}\mathcal{L}_{VQ} + \lambda_f \mathcal{L}_{freq} + \lambda_t \mathcal{L}_{temp}. \quad (2)$$

By incorporating the focal frequency loss and standard reconstruction terms, this unified optimization strategy constructs a detail-rich and stable latent space, laying a robust foundation for the subsequent generative fusion process.

### 3.1.2. STAGE II: STRUCTURE-GUIDED FUSION

Building upon the temporally aligned latent space from Stage I, we aim to synthesize high-fidelity fusion results. We adopt a Decoupled Conditional Adaptation strategy.

We freeze the parameters of the pre-trained 3D-DiT (Song et al., 2025) to strictly preserve its inherent capacity for generating coherent motion dynamics. To bridge the domain gap between the generic video manifold and the specific structural constraints of infrared imagery, we introduce a lightweight Condition Adapter ($\mathcal{P}_{str}$). Unlike simple feature concatenation, this adapter functions as a semantic steering mechanism. It extracts structural primitives $c_{struct}$ from the clean infrared video $I^{IR}$ and projects them into

the 3D-DiT's latent space. The conditioning injection is formulated as:

$$\begin{aligned} \mathbf{c}_{struct} &= \mathcal{P}_{str}(I^{IR}), \\ \mathbf{X}_{in} &= \text{Embed}(\mathbf{z}_k^{VI}) + \mathbf{c}_{struct} + \text{PosEmbed}. \end{aligned} \quad (3)$$

The adapter plays a pivotal role in spectral calibration. By optimizing the adapter, we align the input queries with the frozen attention mechanism, enabling it to effectively distinguish between robust motion trends and high-frequency artifacts even within noisy histories. To ensure numerical stability at the start of optimization, we employ Zero-Initialized Convolutions, initializing the structural guidance as a null signal to smoothly transition from the unconditional prior. The model is optimized using a pixel-level objective that enforces structural fidelity:

$$\mathcal{L}_{total} = \lambda_p \mathcal{L}_{perc} + \lambda_s \mathcal{L}_{ssim} + \lambda_g \mathcal{L}_{grad} + \lambda_i \mathcal{L}_{int} \quad (4)$$

This stage achieves a coarse-grained manifold alignment: the adapter ensures the generated content semantically respects the infrared geometry, while the frozen 3D-DiT maintains temporal consistency. This lays the foundation for the fine-grained latent refinement in the inference phase.

## 3.2. Temporally Stable Inference

During the inference phase, to guarantee both spatial precision and temporal stability, we employ the 3D-DiT as the generative engine, augmented by a gradient-guided refinement strategy and an attention-based anchoring mechanism.

### 3.2.1. LATENT REFINEMENT

While the adapter effectively injects structural information, the domain gap between the latent space and pixel-level edges can occasionally lead to subtle blurring of thermal targets. To address this, we introduce a Latent Refinement strategy.

Applied every $N_{ref}$ sampling steps, this process corrects the predicted latent variable $\hat{\mathbf{z}}_{t,0}$ via posterior projection. We define an energy function $\mathcal{J}$ that maximizes the alignment between the decoded latent and the input modalities:

$$\mathbf{z}^* = \arg \min_{\mathbf{z}} \left( \mathcal{J}(\mathcal{D}(\mathbf{z}), \mathbf{I}_t^{IR}, \mathbf{I}_t^{VI}) + \lambda_{reg}\|\mathbf{z} - \hat{\mathbf{z}}_{t,0}\|_2^2 \right). \quad (5)$$

Subsequently, we update the prediction via a cooperative weighted fusion scheme: $\hat{\mathbf{z}}_{t,0}^{new} = (1 - \gamma) \cdot \hat{\mathbf{z}}_{t,0} + \gamma \cdot \mathbf{z}^*$. Upon completion of the sampling steps ($k \to 0$), the final latent is decoded to yield the fused frame $F_t$.

### 3.2.2. SOFT TEMPORAL ANCHORING

We repurpose the self-attention layers of the 3D-DiT as a Soft Temporal Anchoring mechanism. Unlike naive frame-by-frame generation, we formulate attention as a

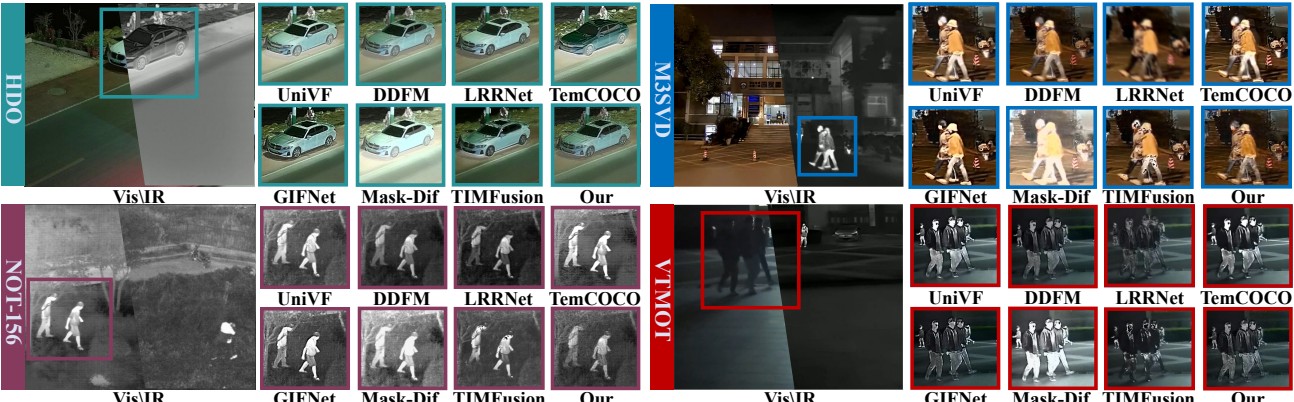

*Figure 3.* Qualitative comparisons between our method and existing fusion approaches on the four datasets.

*Table 1.* Quantitative comparison of IVIF between our method and state-of-the-art methods on HDO, VTMOT, NOT-156, and M3SVD datasets. Red and blue represent the best and second-best results.

| Method | Pub. | HDO | | | M3SVD | | | NOT-156 | | | VTMOT | | |
|---|---|---|---|---|---|---|---|---|---|---|---|---|---|
| | | CC↑ | EN↑ | SSIM↑ | CC↑ | EN↑ | SSIM↑ | CC↑ | EN↑ | SSIM↑ | CC↑ | EN↑ | SSIM↑ |
| CDDFuse | *CVPR'23* | 0.608 | 6.497 | 0.554 | 0.551 | 6.650 | 0.611 | 0.322 | 6.528 | 0.585 | 0.633 | 6.682 | 0.379 |
| DCEvo | *CVPR'25* | 0.656 | 6.349 | 0.573 | 0.584 | 6.452 | 0.630 | 0.301 | 6.274 | 0.605 | 0.568 | 6.637 | 0.439 |
| DDFM | *ICCV'23* | 0.679 | 6.196 | 0.585 | 0.604 | 6.439 | 0.629 | 0.441 | 6.637 | 0.586 | 0.623 | 6.241 | 0.440 |
| GIFNet | *CVPR'25* | 0.656 | 6.303 | 0.579 | 0.543 | 6.533 | 0.628 | 0.413 | 6.129 | 0.599 | 0.622 | 5.824 | 0.315 |
| LRRNet | *TPAMI'23* | 0.646 | 5.917 | 0.591 | 0.474 | 6.561 | 0.606 | 0.445 | 6.418 | 0.535 | 0.636 | 5.715 | 0.349 |
| Mask-Dif | *TPAMI'24* | 0.647 | 6.725 | 0.421 | 0.566 | 7.082 | 0.441 | 0.436 | 6.579 | 0.474 | 0.625 | 7.087 | 0.399 |
| MetaFus | *CVPR'23* | 0.597 | 6.809 | 0.478 | 0.527 | 6.552 | 0.571 | 0.376 | 6.613 | 0.511 | 0.510 | 6.172 | 0.308 |
| PromptF | *JAS'24* | 0.609 | 6.416 | 0.543 | 0.520 | 6.561 | 0.572 | 0.297 | 6.286 | 0.591 | 0.618 | 6.237 | 0.441 |
| SAGE | *CVPR'25* | 0.658 | 6.245 | 0.574 | 0.585 | 6.314 | 0.628 | 0.337 | 6.568 | 0.610 | 0.626 | 6.254 | 0.377 |
| TIMFus | *TPAMI'24* | 0.581 | 5.958 | 0.559 | 0.538 | 6.534 | 0.614 | 0.308 | 6.287 | 0.603 | 0.523 | 5.381 | 0.288 |
| TemCOCO | *ICCV'25* | 0.585 | 6.247 | 0.580 | 0.535 | 6.333 | 0.610 | 0.236 | 6.558 | 0.397 | 0.545 | 6.388 | 0.432 |
| UniVF | *NeurIPS'25* | 0.632 | 6.313 | 0.573 | 0.560 | 6.525 | 0.629 | 0.289 | 6.312 | 0.599 | 0.590 | 6.611 | 0.440 |
| **Ours** | – | **0.682** | 6.692 | **0.595** | 0.594 | 6.574 | **0.635** | **0.458** | **6.664** | **0.612** | **0.639** | 6.783 | **0.450** |

*denoising retrieval process.* For the current frame, we construct a unified attention window $\mathbf{W}_t$ by concatenating the stabilized historical context (detailed in Sec. 3.3) with the current noisy tokens $\mathbf{z}_{t,k}$. The current tokens serve as Queries ($\mathbf{Q}_{t,k}$) to retrieve semantic correspondence from the historical Keys ($\mathbf{K}_{win}$) and Values ($\mathbf{V}_{win}$):
$\mathbf{h}'_{t,k} = \text{Softmax}\left(\frac{\mathbf{Q}_{t,k}\mathbf{K}_{win}^\top}{\sqrt{d_{head}}}\right)\mathbf{V}_{win}$. This mechanism functions as a spectral filter. Unlike rigid optical flow, the attention map acts as a "soft semantic flow." By retrieving exclusively from the stabilized history, it preserves low-frequency motion trends while rejecting high-frequency jitter, effectively breaking the feedback loop of artifact amplification.

## 3.3. Stabilized History Guidance

To implement the soft temporal anchoring described in Sec. 3.2.2, we need to construct a robust retrieval pool that isolates motion trends from accumulated artifacts. We achieve this via a History Modulation mechanism.

### 3.3.1. SPECTRAL-TEMPORAL MODULATION

We apply a noise modulation operator $\mathcal{N}_\lambda(\mathbf{z}) = \alpha_\lambda \mathbf{z} + \sigma_\lambda \boldsymbol{\epsilon}$ to perturb the historical latent variables within the attention window $\mathbf{W}_t$. Note that $\lambda$ denotes the modulation noise level, distinct from the diffusion timestep $k$. We define three parallel historical configurations that effectively modify the Keys ($\mathbf{K}_{win}$) and Values ($\mathbf{V}_{win}$) for the attention layer:

**Baseline History ($\mathbf{H}^{(0)}$).** All historical frames in $\mathbf{W}_t$ are replaced with pure Gaussian noise. This effectively "blinds" the attention mechanism to temporal context, forcing the model to rely solely on the structural conditions of the current frame (i.e., the Adapter output $\mathbf{c}_{struct}$ and noisy input $\mathbf{z}_t^{VI}$). This serves as the temporal-unconditional baseline, equivalent to the single-frame fusion in Stage II.

**Stabilized History ($\mathbf{H}^{(1)}$).** Historical frames are perturbed with minimal noise. As derived by Song et al. (Song et al., 2025), this fractional noise injection acts as a soft low-pass filter in the frequency domain. It allows the attention mechanism to capture global motion semantics (low-frequency) while suppressing pixel-level jitter stored in the history

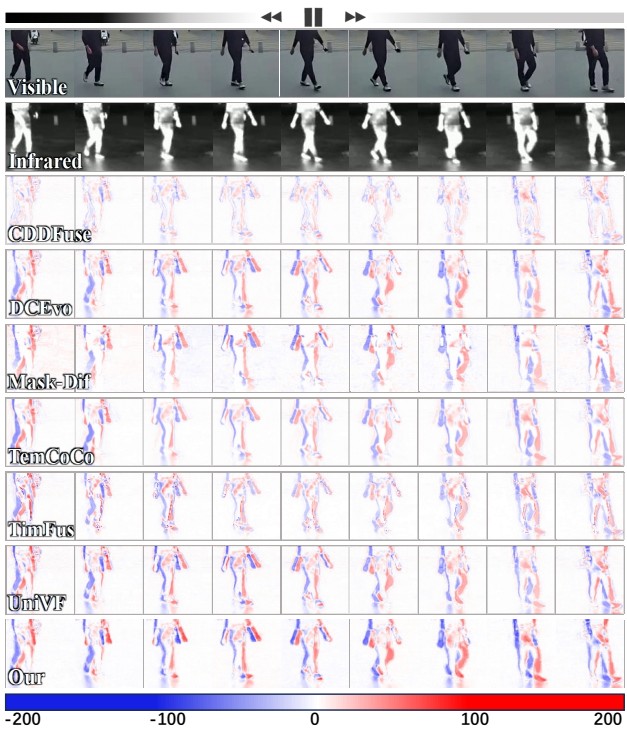

*Figure 4.* Frame-wise difference visualization on the M3SVD dataset for evaluating temporal consistency.

*Table 2.* Quantitative comparison on HDO, M3SVD, NOT-156, and VTMOT datasets. The table is split into two parts for better readability. Red and blue indicate the best and second-best.

| Method | HDO | | M3SVD | |
|---|---|---|---|---|
| | BiSWE↓ | MS2R↓ | BiSWE↓ | MS2R↓ |
| CDDFuse | 7.398 | 0.220 | 8.882 | 0.238 |
| DCEvo | 6.335 | 0.223 | 7.395 | 0.227 |
| DDFM | 6.544 | 0.232 | 6.676 | 0.228 |
| GIFNet | 7.257 | 0.220 | 7.657 | 0.235 |
| LRRNet | **5.954** | 0.231 | 7.201 | 0.310 |
| Mask-Dif | 8.203 | **0.211** | 9.877 | 0.228 |
| MetaFus | 12.259 | 0.224 | 12.556 | 0.251 |
| PromptF | 6.333 | 0.219 | 8.366 | 0.233 |
| SAGE | 8.680 | 0.219 | 7.207 | 0.229 |
| TIMFus | 6.565 | 0.214 | 6.926 | 0.232 |
| TemCOCO | 6.414 | 0.213 | 6.488 | 0.258 |
| UniVF | 6.378 | 0.227 | 7.316 | 0.222 |
| **Ours** | 6.225 | **0.211** | **6.467** | **0.220** |

| Method | NOT-156 | | VTMOT | |
|---|---|---|---|---|
| | BiSWE↓ | MS2R↓ | BiSWE↓ | MS2R↓ |
| CDDFuse | 6.358 | 0.374 | 9.983 | 0.588 |
| DCEvo | 4.848 | 0.345 | 8.579 | 0.577 |
| DDFM | 5.221 | 0.367 | 7.992 | 0.621 |
| GIFNet | 5.447 | 0.347 | 10.239 | 0.600 |
| LRRNet | 5.494 | 0.339 | 8.831 | 0.616 |
| Mask-Dif | 6.089 | 0.343 | 10.300 | 0.611 |
| MetaFus | 8.327 | 0.421 | 15.597 | 0.609 |
| PromptF | 6.276 | 0.352 | 8.348 | 0.595 |
| SAGE | 4.871 | 0.335 | 8.548 | 0.626 |
| TIMFus | 6.370 | 0.334 | 7.618 | 0.589 |
| TemCOCO | **4.593** | 0.382 | 8.122 | 0.826 |
| UniVF | 4.690 | 0.338 | 8.551 | 0.591 |
| **Ours** | 4.816 | **0.332** | **7.418** | **0.560** |

(high-frequency), serving as the primary motion reference.

**Context Suppression ($\mathbf{H}^{(2)}$).** Only the most recent previous frame is retained, while distant history is masked out. This configuration captures the strongest short-term dependencies, which often contain the most immediate flickering or registration errors.

### 3.3.2. GUIDANCE COMPOSITION

These three configurations are processed in parallel via the frozen 3D-DiT, while keeping the adapter's structural condition $\mathbf{c}_{struct}$ constant. Since the attention mechanism aggregates features from each history differently, this yields three distinct velocity estimates. We synthesize the final guided velocity $\mathbf{v}_{guided}$ through a linear combination:

$$\mathbf{v}_{guided} = \mathbf{v}(\mathbf{z}_{t,k}|\mathbf{H}^{(0)}) + s \cdot \left[\mathbf{v}(\mathbf{z}_{t,k}|\mathbf{H}^{(1)}) - \mathbf{v}(\mathbf{z}_{t,k}|\mathbf{H}^{(2)})\right],$$
(6)

where $s$ denotes the guidance scale. Leveraging this "Cross-Time and Frequency" strategy, we effectively separate robust motion signals from local noise. By subtracting the short-term dependency captured in $\mathbf{v}(\cdot|\mathbf{H}^{(2)})$ from the spectrally stabilized prediction $\mathbf{v}(\cdot|\mathbf{H}^{(1)})$, we derive a pure long-range temporal gradient that remains invariant to local inter-frame fluctuations.

## 4. Experiments

In this section, we conduct comprehensive experiments to evaluate the effectiveness and efficiency of the proposed DRFusion framework. We first detail the experimental settings and implementation. Then, we present extensive quantitative and qualitative comparisons against state-of-the-art methods on four benchmark datasets (HDO (Xie et al., 2024a), M3SVD (Tang et al., 2025b), NOT-156 (Sun et al., 2025), VTMOT (Zhao et al., 2025)) in terms of fusion quality, temporal consistency, and utility in downstream visual tasks. To validate the contribution of each key design, we perform a thorough ablation study. Finally, we analyze the inference efficiency, demonstrating the practical viability of our approach.

### 4.1. Set up

**Dataset.** We conduct comprehensive evaluations on four infrared and visible video fusion benchmarks: HDO, M3SVD, NOT-156, and VTMOT. These datasets cover diverse environments and motion dynamics, enabling a comprehensive

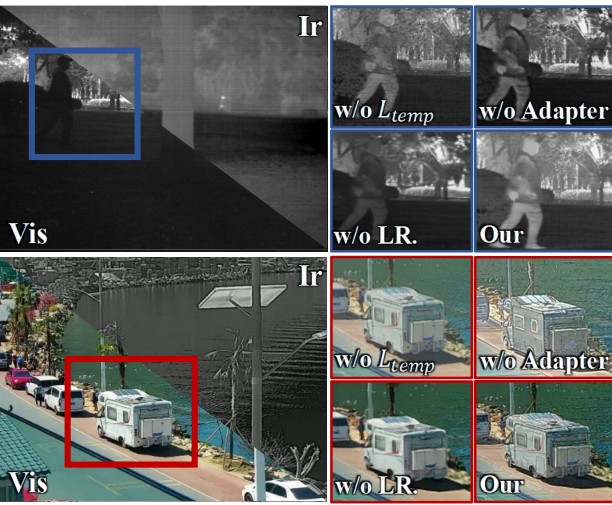

*Figure 5.* Qualitative comparison of different ablation variants on the HDO and NOT-156 datasets.

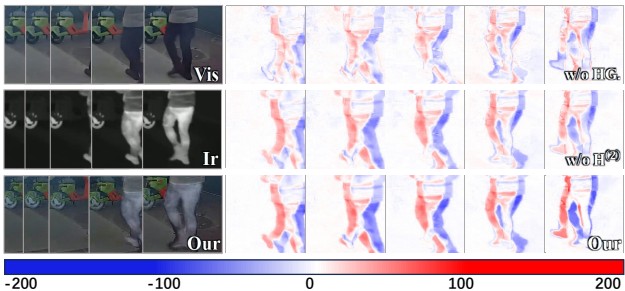

*Figure 6.* Frame-wise difference visualization of ablation variants on the M3SVD dataset for evaluating temporal consistency.

evaluation of both fusion quality and temporal consistency.

**Implementation Details.** The batch size in Stage I is set to 16 for 50 epochs. In Stage II, we train for 100 epochs with a batch size of 2. The network parameters are optimized using AdamW with a learning rate of $1 \times 10^{-4}$. During the inference phase, we employ the DDIM sampler with 50 sampling steps for efficient generation. For the proposed mechanisms, the sliding history window length is set to $T = 8$. The Stabilized History Guidance is configured with a guidance scale of $s = 2.0$ and a noise injection level of $\sigma_{stab} = 0.02$. The Latent Refinement is applied every 5 steps ($N_{ref} = 5$). All experiments are conducted on two NVIDIA A40 GPUs.

## 4.2. Results of multi-modality video fusion

In this section, we compared the proposed method with 12 state-of-the-art fusion methods, including CDDFuse (Zhao et al., 2023b), DCEVO (Liu et al., 2025a), DDFM (Zhao et al., 2023c), GIFNet (Cheng et al., 2025), Mask-Dif (Tang et al., 2025a), MetaFusion (Zhao et al., 2023a), PromptFusion (Liu et al., 2024a), SAGE (Wu et al., 2025), TIMFu-

sion (Liu et al., 2024b), LRRNet (Li et al., 2023a),as well as video-based methods TemCOCO (Gong et al., 2025) and UniVF (Zhao et al., 2025).

*Table 3.* Quantitative results of the ablation study on the NOT-156 dataset. The best results are in **bold**

|  | CC↑ | EN↑ | SSIM↑ | BiSWE↓ | MS2R↓ |
|---|---|---|---|---|---|
| w/o $\mathcal{L}_{temp}$ | 0.442 | 6.523 | 0.589 | 5.172 | 0.358 |
| w/o Adapter | 0.432 | 6.587 | 0.544 | 5.214 | 0.342 |
| w/o LR. | 0.328 | 5.932 | 0.547 | 4.911 | 0.342 |
| w/o HG. | 0.438 | 6.487 | 0.571 | 5.238 | 0.361 |
| w/o $\mathbf{H}^{(2)}$ | 0.440 | 6.501 | 0.601 | 4.972 | 0.341 |
| **Ours** | **0.458** | **6.664** | **0.612** | **4.816** | **0.332** |

**Quantitative Comparisons.** We evaluate the fusion performance using three objective metrics: Correlation Coefficient (CC), Entropy (EN), and Structural Similarity (SSIM). Table 1 reports the quantitative results, where red and blue indicate the best and second-best performance respectively. As observed, our method achieves the best results in the most metrics. In particular, our method ranks first in SSIM across all four datasets, demonstrating its superior capability to preserve structural information from source scenes. Additionally, we achieve the highest CC scores on three datasets (HDO, NOT-156, and VTMOT) and the second-highest on M3SVD, indicating that our fused images maintain the highest correlation with the source modalities. While some methods like Mask-Dif perform well on EN, our method maintains a better balance between information richness and structural fidelity.

**Qualitative Comparisons.** To intuitively analyze the fusion quality, we select representative frames from the four datasets for visual comparison, as shown in Figure 3. Distinct from other methods that may introduce blurring or ghosting artifacts, our method excels in structural restoration and detail sharpness. As observed in the zoomed-in regions, our method effectively preserves the clean edges of thermal targets, such as the car contours in HDO and pedestrians in M3SVD, while maintaining the fine textures of the background. The superior structural clarity aligns with our quantitative advantage in SSIM, indicating that our method produces more natural and accurate fused images.

## 4.3. Evaluation of temporal consistency

To verify the temporal stability and continuity of the fused video, we visualize the pixel-wise intensity changes between adjacent frames on a sequence from the M3SVD dataset. Specifically, we calculate the difference map by subtracting the previous frame from the current one, where positive deviations are mapped to red and negative ones to blue. In an ideal temporally consistent video, static background regions should remain pure white (indicating zero fluctuation), while moving objects should exhibit clear and continuous

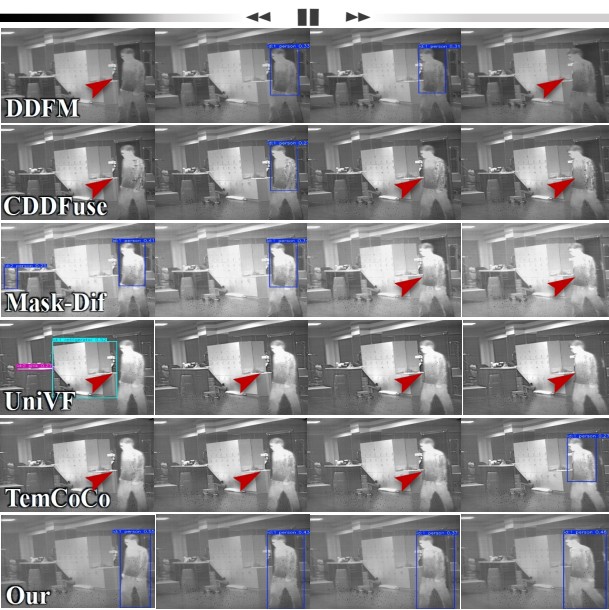

*Figure 7.* Visual comparison of object track on NOT-156.

duces clearer thermal structures, sharper textures, and more coherent frame-wise motion than the ablated variants.

*Table 4.* Quantitative comparison of object tracking performance on the NOT-156 dataset. Red and blue represent the best and second-best results.

| Method | AUC↑ | DP@20 ↑ | SR@0.5 ↑ | SR@0.75 ↑ |
|--------|------|---------|----------|-----------|
| CDDFuse | 0.220 | 0.209 | 0.208 | 0.084 |
| DCEvo | 0.221 | 0.211 | 0.189 | 0.082 |
| DDFM | 0.240 | 0.215 | 0.221 | 0.090 |
| GIFNet | 0.224 | 0.225 | 0.216 | 0.089 |
| LRRNet | 0.223 | **0.231** | 0.217 | 0.108 |
| Mask-Dif | 0.206 | 0.203 | 0.197 | 0.100 |
| MetaFus | 0.231 | 0.228 | 0.205 | 0.099 |
| PromptF | 0.197 | 0.186 | 0.177 | 0.081 |
| SAGE | 0.216 | 0.200 | 0.205 | 0.088 |
| TIMFus | 0.195 | 0.166 | 0.182 | 0.095 |
| TemCOCO | 0.185 | 0.187 | 0.150 | 0.077 |
| UniVF | 0.212 | 0.217 | 0.196 | 0.071 |
| **Ours** | **0.252** | 0.230 | **0.263** | **0.121** |

### 4.5. Object tracking

To evaluate the downstream utility of the fused videos, we conduct object tracking experiments on the NOT-156 dataset. We use ByteTrack with a pre-trained YOLOv11n detector without fine-tuning, and report AUC, Distance Precision (DP), and Success Rate (SR). As shown in Table 4, our method achieves the best performance on most metrics, including AUC and both SR thresholds. The improvement on the stricter SR@0.75 metric indicates that our fused videos support more accurate target localization.

Figure 7 shows a challenging sequence where the target person is mixed with a cluttered background. Compared with competing methods that suffer from missed detections or unstable tracking, our method maintains more continuous target responses, demonstrating that DRFusion provides more reliable fused inputs for downstream video perception.

## 5. Conclusion

We propose DRFusion, a drift-resilient infrared-visible video fusion method that reformulates the task as a history-conditioned motion generation process. By introducing the Stabilized History Guidance mechanism and a Decoupled Structure-Motion Adaptation strategy, DRFusion treats temporal consistency as a spectral filtering problem. This design effectively isolates robust low-frequency motion trends from high-frequency error accumulation, enabling drift-free generation without the geometric rigidity of optical flow. Extensive experiments on diverse video fusion datasets validate the superiority of our approach in both fusion quality and temporal stability.

color trajectories. As shown in Figure 4, our method yields cleaner backgrounds and sharper motion trails compared to others (e.g., Mask-Dif shows scattered noise), demonstrating superior temporal coherence.

We further quantify temporal stability employing two video quality metrics: BiSWE and MS2RF (Zhao et al., 2025), where lower values (↓) indicate better temporal coherence. As reported in Table 2, our method achieves the best performance (marked in red) in the HDO and VTMOT datasets for both metrics, and ranks first or second on M3SVD and NOT-156. These consistent low scores across different scenarios confirm that our proposed framework effectively minimizes frame-to-frame fluctuations, ensuring robust temporal consistency for video fusion tasks.

### 4.4. Ablation study

To investigate the contribution of each key component, we conduct ablation studies by removing the occlusion-aware temporal loss $\mathcal{L}_{temp}$ in Stage I, disabling the trained Condition Adapter in Stage II, omitting Latent Refinement (LR), replacing History Guidance (HG.) with a vanilla autoregressive setting, and removing the context suppression term $H^{(2)}$.

Table 3 reports the quantitative results on the NOT-156 dataset. Removing any component leads to clear degradation in either fusion quality or temporal stability, confirming that these modules are complementary. In particular, removing HG. causes unstable temporal variations, while disabling the Condition Adapter or LR weakens structural preservation and target clarity. The visual comparisons in Figure 5 and Figure 6 further show that the full model pro-

## Acknowledgments

This work was partially supported by the China Postdoctoral Science Foundation under Grant 2023M730741, the National Natural Science Foundation of China under Grants 62302078 and 62372080, and the Key Research and Development Program of Liaoning Province under Grant 2023JH26/10200014.

## Impact Statement

DRFusion significantly enhances video perception in low-light and dynamic environments, benefiting critical applications such as autonomous driving, all-weather surveillance, and search and rescue operations. While this work has a broad range of potential societal consequences, we believe there are no specific negative impacts or unique ethical risks that must be highlighted here.

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
