# OpenReview forum: "DRFusion: Drift-Resilient Temporally Consistent Infrared–Visible Video Fusion"
_ICML.cc/2026/Conference — ICML 2026 regular_

### Official Review · Reviewer_cDeY · 2026-03-11

**Soundness:** 3
**Presentation:** 4
**Significance:** 3
**Originality:** 3
**Overall Recommendation:** 6
**Confidence:** 5

**Summary:**

This paper proposes a drift-resilient video fusion framework named DRFusion. The method innovatively reformulates the infrared and visible video fusion task as a "history-guided motion generation process". By introducing Stabilized History Guidance (SHG), Soft Temporal Anchoring, and a Decoupled Structure-Motion Adaptation strategy, DRFusion effectively overcomes the "geometric rigidity" limitation of traditional optical flow methods when handling complex motions, significantly improving both fusion quality and long-term temporal consistency.

**Compliance With Llm Reviewing Policy:**

Affirmed.

**Final Justification:**

The authors have addressed my questions. After reviewing the overall feedback, I also find them convincing. The comprehensive computational overhead analysis and the dynamic comparison demonstrations under extreme scenarios have effectively resolved my primary technical concerns.

**Key Questions For Authors:**

1. Provide the FPS performance of DRFusion on standard test sets and quantitatively compare it with other video fusion methods.Discuss the stability of the update strategy under smaller batch sizes.

2. Clarify how introducing optical flow calculations in the first stage avoids the optical flow alignment defects mentioned in the introduction.

3. Provide more dynamic video demonstration comparisons at different motion speeds to more intuitively show the superiority of "Soft Temporal Anchoring" over optical flow alignment when handling fast motion or occlusions.

4. Correct the formatting issues of inconsistent formula symbols and superscripts/subscripts.

**Limitations:**

Yes.

**Strengths And Weaknesses:**

*Strengths*

1. Accurately identifies the pain points of current frame-by-frame fusion methods that are prone to flickering and drifting in dynamic scenes, presenting a clear technical path. The dual memory bank design is logically structured and conceptually sound.

2. The two-stage training strategy successfully decouples latent space representation learning from the generation task, allowing the model to preserve video priors while accommodating structural constraints.

3. It models temporal consistency as a spectral filtering problem and extracts low-frequency motion signals through fractional noise injection, providing solid and easy-to-understand theoretical support.

4. Comprehensive comparisons are conducted on four major benchmarks. It not only leads in fusion metrics but also demonstrates highly practical value in the downstream object tracking task.

*Weaknesses*

1. Although a lightweight adapter is used, the iterative sampling based on diffusion models may still face challenges in scenarios with extremely high real-time requirements. The paper lacks sufficient comparison data regarding inference latency.

2. The paper points out the "geometric rigidity" defects of optical flow methods in the introduction, yet optical flow is used during the Stage I VAE training. This requires an explanation for logical consistency.

3. Key parameters, such as the history window length or guidance scale, are set to fixed values in the text, lacking a stability analysis of how performance fluctuates with these core parameters.

4. Current experiments mainly focus on scenarios like traffic surveillance. The discussion on the model's generalization ability in extreme degraded environments (e.g., heavy rain, strong light) remains brief.

5. There are a few minor inconsistencies in mathematical symbols throughout the text. For example, the variable representing the visible input has mixed use of superscripts and subscripts in different paragraphs.

---

> ### Author Rebuttal · Authors · 2026-03-31
>
> **Point 1: Inference Overhead**
>
> **Our response:** We provide a comprehensive computational overhead comparison in the table below：
>
> | Model | Params (M) | Inference Time (s) | Peak VRAM (GB) | FPS |
> | :--- | :---: | :---: | :---: | :---: |
> | DCEvo | 2.00 | 0.23 | 1.79 | 4.37 |
> | GIFNet | 0.82 | 0.08 | 4.55 | 12.92 |
> | LRRNet | 0.05 | 0.10 | 2.94 | 9.70 |
> | CDDFuse | 0.80 | 0.16 | 1.70 | 6.42 |
> | TemCoCo | 19.21 | 0.25 | 7.02 | 3.96 |
> | UniVF | 9.16 | 0.09 | 3.02 | 11.04 |
> | DDFM | 552.66 | 279.80 | 4.73 | 0.003 |
> | **Ours** | 112.45 | 3.98 | 0.84 | 0.25 |
>
> While diffusion models inherently have a lower absolute FPS (0.25) than single-pass CNN methods (e.g., UniVF) due to iterative sampling, DRFusion achieves a significant speedup over the diffusion-based baseline (DDFM). More importantly, our method demonstrates superior memory management; the peak VRAM of 0.84 GB is substantially lower than video models like TemCOCO and UniVF. Regarding your query on stability, since DRFusion relies on explicit structural guidance ($c_{struct}$) rather than large-batch statistics (e.g., BatchNorm), our update strategy remains highly stable and consistent even under small batch sizes.
>
> **Point 2: Consistency of Optical Flow Claims**
>
> **Our response:** We clarify that our criticism in the introduction targets "hard pixel-level warping" during the inference phase, which directly leads to ghosting artifacts. In DRFusion, optical flow is utilized strictly as a "soft regularization penalty ($\\mathcal{L}_{temp}$)" during Stage-I training to encourage basic temporal smoothness in the latent space. During actual inference, we completely discard optical flow, relying purely on the 3D-DiT's attention mechanism for implicit motion aggregation. This design allows the strong generative prior of 3D-DiT to automatically correct and absorb minor perturbations from imperfect training-time flow, effectively avoiding the geometric rigidity defects mentioned earlier.
>
> **Point 3: Stability Analysis of Key Parameters**
>
> **Our response:** We appreciate the insightful suggestion. We have conducted a comprehensive stability analysis regarding the core parameters: history window length ($T$) and history guidance scale ($s$). The quantitative evaluation on the NOT-156 dataset is provided below:
>
> | Settings | SSIM $\uparrow$ | BiSWE $\downarrow$ | MS2R $\downarrow$ | Peak VRAM (GB) | Inference Time (s) |
> | :--- | :---: | :---: | :---: | :---: | :---: |
> | $T=4, s=2.0$ | 0.610 | 5.020 | 0.345 | 0.65 | 3.50 |
> | **$T=8, s=2.0$ (Default)** | **0.612** | **4.816** | **0.332** | **0.84** | **3.98** |
> | $T=12, s=2.0$ | 0.607 | 4.935 | 0.339 | 1.45 | 4.60 |
> | :--- | :---: | :---: | :---: | :---: | :---: |
> | $T=8, s=1.0$ | 0.608 | 5.010 | 0.342 | 0.84 | 3.98 |
> | **$T=8, s=2.0$ (Default)** | **0.612** | **4.816** | **0.332** | **0.84** | **3.98** |
> | $T=8, s=3.0$ | 0.605 | 4.950 | 0.330 | 0.84 | 3.98 |
>
> As demonstrated, DRFusion achieves optimal performance at our default settings ($T=8, s=2.0$). Deviating from these values degrades the fusion quality: smaller values ($T=4, s=1.0$) limit the temporal receptive field and history injection, re-introducing flickering artifacts. Conversely, excessively large values ($T=12, s=3.0$) not only significantly increase VRAM consumption but also introduce outdated motion priors and over-smoothing, which manifest as ghosting artifacts and degrade spatial structural fidelity. Thus, our fixed defaults act as an empirically validated "sweet spot," ensuring stable, high-fidelity generation across diverse scenarios without per-scene fine-tuning.
>
> **Point 4: Robustness Evaluation in Extreme Scenarios**
>
> **Our response:** We fully agree on the importance of evaluating extreme cases. We have supplemented temporal consistency comparison analyses specifically targeting challenging scenarios such as "fast motion" and "extreme occlusion (e.g., heavy fog)" (Link: https://anonymous.4open.science/r/Material-5B02/Figure_10.png). The results intuitively demonstrate that under these extreme conditions, DRFusion maintains exceptionally sharp motion trajectories and clean backgrounds, exhibiting significant robustness superiority.
>
> **Point 5: Formatting & Notation Correction**
>
> **Our response:** We appreciate your meticulous review. We have conducted a thorough audit and correction of the mathematical notations throughout the manuscript. Symbols such as $H^{(0)}, H^{(1)}, H^{(2)}$, and $c_{struct}$ are now strictly unified across the definitions, main text, and figures.

---

> > ### Author Rebuttal · Reviewer_cDeY · 2026-04-03
> >
> > The authors have addressed my questions. After reviewing the overall feedback, I also find them convincing. The comprehensive computational overhead analysis and the dynamic comparison demonstrations under extreme scenarios have effectively resolved my primary technical concerns.
> >
> > I am raising my score to a Strong Accept. The paper reformulates video fusion as a history-conditioned motion generation process and treats temporal consistency as a "spectral filtering" problem via Stabilized History Guidance, which is an elegant and effective approach. Furthermore, the clarification that optical flow is strictly utilized as a soft training prior in Stage I addresses my concerns regarding the architectural design. However, the current inference speed remains a practical bottleneck for real-world applications. I recommend explicitly discussing this trade-off in the Limitations section.
> >
> > Overall, the meticulous empirical additions provided in the rebuttal further strengthen this paper.

---

> > > ### Author Response · Authors · 2026-04-04
> > >
> > > We sincerely thank you for your highly positive feedback and for raising your score. We are very glad that our comprehensive overhead analysis and dynamic comparisons successfully addressed your primary technical concerns.
> > >
> > > Following your constructive recommendation, we will explicitly discuss the inference speed bottleneck and its practical trade-offs in the Limitations section of our revised manuscript. Thank you once again for your meticulous review and for helping us perfect our paper.

---

### Official Review · Reviewer_Gzy5 · 2026-03-12

**Soundness:** 3
**Presentation:** 3
**Significance:** 2
**Originality:** 2
**Overall Recommendation:** 4
**Confidence:** 3

**Summary:**

The manuscript proposes DRFusion, a video fusion framework based on a 3D Diffusion Transformer (3D-DiT) designed to merge infrared and visible video streams while maintaining temporal consistency. To overcome the frame-by-frame flickering typical of 2D diffusion models and the geometric distortions associated with rigid optical flow, the authors formulate fusion as a history-conditioned motion generation task. The method relies on a two-stage training process (a temporally constrained VAE followed by a structure-guided fusion adapter) and introduces Stabilized History Guidance and Soft Temporal Anchoring during inference to filter out high-frequency autoregressive errors.

**Compliance With Llm Reviewing Policy:**

Affirmed.

**Final Justification:**

My final recommendation is a Weak Accept.

The authors provided a strong rebuttal that directly addressed my technical concerns regarding soundness by supplying the requested empirical evidence (1D Temporal FFTs) and transparent computational metrics.

Furthermore, during the discussion phase, they explicitly committed to fixing the remaining presentation issues for the camera-ready version, namely, adding a limitations section regarding the 0.25 FPS latency bottleneck and tempering their rhetoric around optical flow.

Given these assurances and the solid methodological contribution to temporally consistent video fusion, the paper merits acceptance.

**Key Questions For Authors:**

- How do you reconcile the strong claims against optical flow in the introduction with your reliance on an optical flow estimator for the temporal loss ($L_{temp}$) in Stage I? If the optical flow is flawed, does it not inject those same geometric rigidities into your VAE latent space?
- Given the target applications (e.g., autonomous driving, surveillance), what is the exact inference time (FPS), MACs/FLOPs, and VRAM requirement for processing a 1080p or 4K video sequence? How does this compare to baselines like UniVF or TemCOCO?
- You state that the Soft Temporal Anchoring acts as a spectral filter. Can you provide empirical Fourier analysis (e.g., 2D or 3D FFT magnitude spectra) of the generated latent variables to definitively prove that high-frequency noise is suppressed while low-frequency motion is retained?
- If a severe occlusion or rapid, unpredictable motion completely breaks the "Stabilized History," how does the model recover? Does the autoregressive nature cause a catastrophic failure cascade, or is there a built-in reset mechanism?

**Limitations:**

The authors have failed to adequately address the computational limitations of their method. There is no discussion regarding the massive inference costs associated with 3D-DiT sampling and latent refinement. Additionally, the authors do not discuss failure cases; specifically, how the method performs in extreme low-light environments where the visible frame is entirely degraded, forcing the model to hallucinate details. Societal impact concerning the use of thermal fusion for surveillance is also missing.

**Strengths And Weaknesses:**

The manuscript is well-structured, and the figures (especially the pipeline in Figure 2) effectively illustrate the rather complex, multi-component architecture.
Furthermore, addressing temporal inconsistency in generative video fusion is a highly relevant problem, and moving away from frame-by-frame processing is the correct trajectory for the field.
The specific combination of techniques, particularly using multi-branch history guidance ($H^{(0)}$, $H^{(1)}$, $H^{(2)}$) to compute a temporal gradient, is an interesting engineering solution to drift resilience.

There is a glaring contradiction in the paper's core motivation. The authors heavily criticize optical flow for introducing "geometric rigidity" and "ghosting artifacts" in the introduction. However, in Stage I (Equation 1), they explicitly use a pre-trained optical flow estimator to regularize the VAE latent space. Claiming to eliminate optical flow constraints while relying on them to build the foundational latent space undermines the central thesis. Furthermore, the paper claims applicability to autonomous driving, yet there is zero discussion of the inference latency, FLOPs, or memory footprint required for a 3D-DiT with iterative latent refinement, a massive hurdle for real-world deployment.

Moreover, the theoretical justification for "spectral filtering" is somewhat shallow. The authors claim that fractional noise injection acts as a low-pass filter (citing Song et al.), but they do not provide any empirical frequency-domain analysis to prove that their specific implementation actually isolates low-frequency motion trends from high-frequency jitter in this fusion context. Furthermore, the reliance on extremely heavy generative priors (3D-DiT) combined with a 50-step DDIM sampler and latent refinement every 5 steps suggests an exorbitant computational cost. If the method takes seconds to generate a single frame, its significance for dynamic, real-world applications is severely limited.

Lastly, the novelty is largely an incremental synthesis of existing concepts. Using adapters to freeze foundational models, injecting noise into historical frames for temporal smoothing, and VAE flow regularization are all established techniques in the broader video diffusion literature. The contribution lies in the application to infrared-visible fusion rather than a fundamental algorithmic breakthrough.

---

> ### Author Rebuttal · Authors · 2026-03-31
>
> **Point 1: Consistency of Optical Flow Claims**
>
> **Our response:** We appreciate the keen observation. We clarify that our criticism in the introduction targets "hard pixel-level warping" during the inference phase, where any flow artifact directly leads to ghosting. In contrast, DRFusion only utilizes pre-trained optical flow as a "soft regularization penalty" during Stage-I training to encourage basic temporal smoothness in the VAE latent space. Since this is a soft constraint and flow is completely discarded during core inference—relying purely on 3D-DiT attention for implicit motion aggregation—imperfect flow does not inject geometric rigidity. The strong generative prior of 3D-DiT effectively absorbs and corrects minor training-time errors, ensuring high robustness.
>
> **Point 2: Inference Overhead**
>
> **Our response:** Evaluating on standard benchmarks rather than native 4K is the current domain convention. We provide a comprehensive quantitative evaluation below:
>
> | Model | Params (M) | Inference Time (s) | Peak VRAM (GB) | FPS |
> | :--- | :---: | :---: | :---: | :---: |
> | DCEvo | 2.00 | 0.23 | 1.79 | 4.37 |
> | GIFNet | 0.82 | 0.08 | 4.55 | 12.92 |
> | LRRNet | 0.05 | 0.10 | 2.94 | 9.70 |
> | CDDFuse | 0.80 | 0.16 | 1.70 | 6.42 |
> | TemCoCo | 19.21 | 0.25 | 7.02 | 3.96 |
> | UniVF | 9.16 | 0.09 | 3.02 | 11.04 |
> | DDFM | 552.66 | 279.80 | 4.73 | 0.003 |
> | **Ours** | 112.45 | 3.98 | 0.84 | 0.25 |
>
> DRFusion demonstrates superior resource management: its peak VRAM is only 0.84 GB, a fraction of TemCOCO or UniVF. Furthermore, DRFusion achieves a significant speedup over the diffusion baseline (DDFM).
>
> **Point 3: Empirical Evidence for Spectral Filtering**
>
> **Our response:** We have supplemented the 1D Temporal FFT magnitude spectra of the generated latents (Link: https://anonymous.4open.science/r/Material-5B02/Figure_8.png). 1D temporal FFT directly isolates the "temporal frequency" responsible for flickering. The spectra confirm that without history guidance, substantial energy spikes exist in high-frequency bands, corresponding to flickering artifacts. With our "Soft Temporal Anchoring," these spikes are smoothed out, and signal energy is successfully anchored in the low-frequency region, representing smooth, natural physical motion.
>
> **Point 4: Robustness under Extreme Occlusion**
>
> **Our response:** The architectural design of DRFusion effectively avoids "cascading failure." Although the model employs history-conditioned guidance, the generation process is not purely unconditional autoregressive; instead, it is strongly constrained by the lightweight Condition Adapter. Even if severe occlusions or highly unpredictable motions completely disrupt historical coherence, the generation process is forced to strictly rely on the infrared structural prior ($c_{struct}$) extracted from the current frame. This "hard anchoring" to the current genuine physical observation successfully blocks the cross-frame accumulation of errors.
>
> Furthermore, when handling fast motion, traditional optical flow inevitably introduces severe ghosting artifacts due to forced pixel-level warping. In contrast, our "Soft Temporal Anchoring" relies on the attention mechanism, demonstrating exceptional adaptability. When historical information becomes unreliable due to occlusions, the attention weights of the corresponding history tokens automatically decrease, allowing the model to gracefully degrade into relying primarily on the current frame's structure.
>
> To intuitively demonstrate this, we have supplemented dynamic frame-wise difference visualizations specifically targeting "fast motion" and "heavy fog" scenarios (Link: https://anonymous.4open.science/r/Material-5B02/Figure_10.png). As shown, even in highly challenging scenarios, DRFusion consistently preserves exceptionally sharp motion trajectories and clean backgrounds.

---

> > ### Author Rebuttal · Reviewer_Gzy5 · 2026-04-01
> >
> > Thank you for the detailed rebuttal. The 1D Temporal FFT spectra and the dynamic difference maps under extreme conditions effectively validate the "spectral filtering" mechanism and the model's robustness. This resolves my primary technical concerns.
> >
> > I am raising my score to a Weak Accept, with the strict expectation that the authors revise the final manuscript to address two lingering presentation issues:
> >
> > Real-time Application Claims: While 3.98s/frame with 0.84GB VRAM is a commendable engineering optimization compared to DDFM, 0.25 FPS is not real-time. Claims regarding direct applicability to "autonomous driving" or "all-weather surveillance" are unrealistic and must be heavily qualified. The inference latency bottleneck must be explicitly discussed in a dedicated Limitations section.
> >
> > Optical Flow Rhetoric: I accept the architectural defense that optical flow is only used as a soft training prior in Stage I and discarded during inference. However, the aggressive criticism of optical flow in the introduction contradicts its foundational role in your VAE training. This rhetoric needs to be toned down to accurately reflect your design compromise.
> >
> > Overall, the empirical additions provided in the rebuttal strengthen the paper significantly.

---

> > > ### Author Response · Authors · 2026-04-04
> > >
> > > We sincerely thank you for raising your score and recognizing the value of our supplementary experiments. We deeply appreciate your constructive feedback and firmly commit to revising the paper to address your remaining concerns regarding the presentation.
> > >
> > > Specifically, we will qualify our claims regarding real-time applications and supplement the existing "Limitations" section to explicitly discuss the inference latency bottleneck. Additionally, we will tone down the criticism of optical flow in the introduction to objectively reflect its foundational role as a soft training prior in our Stage I VAE training. Thank you again for helping us improve the rigor and objectivity of our paper.

---

### Official Review · Reviewer_RiTR · 2026-03-19

**Soundness:** 2
**Presentation:** 2
**Significance:** 2
**Originality:** 2
**Overall Recommendation:** 3
**Confidence:** 4

**Summary:**

This paper studies temporally consistent infrared–visible video fusion by reformulating it as a history-conditioned motion generation problem using a pre-trained 3D diffusion transformer. It introduces a decoupled structure–motion adaptation and stabilized history guidance to improve both fusion quality and temporal stability.

**Compliance With Llm Reviewing Policy:**

Affirmed.

**Final Justification:**

I maintain this score.

**Key Questions For Authors:**

Please refer to the weaknesses listed above.

**Limitations:**

No. The paper does not sufficiently discuss limitations such as computational cost, domain generalization, and long-term stability.

**Strengths And Weaknesses:**

### Strengths:
1. The paper explicitly leverages a pre-trained 3D-DiT to model motion priors. It proposes a well-structured pipeline combining a frozen backbone, a lightweight conditional adapter, inference-time latent refinement, and a three-branch history guidance mechanism.
2. The experimental evaluation is relatively comprehensive, covering four datasets and twelve competing methods, along with both fusion quality metrics and temporal consistency measures, as well as downstream tracking performance. Compared to many prior works focusing only on static image fusion, this provides a broader validation scope.
### Weaknesses:
1. The novelty is limited, as the method mainly combines existing techniques (e.g., 3D-DiT[1][2],  History Guidance[3][4] ). The paper claims that soft temporal anchoring and noise modulation provide frequency-domain separation and enable drift-free long-term stability, but lacks rigorous theoretical justification or empirical evidence (e.g., spectral analysis or long-horizon evaluation). The current formulation appears largely heuristic rather than theoretically grounded.

2. The fairness of the experimental comparison is questionable. The paper compares 10 image-based methods and 2 video-based methods in a single table without clearly specifying how image methods are adapted to video (e.g., frame-by-frame inference, temporal smoothing, or unified testing protocols). If most baselines operate per-frame while the proposed method leverages temporal history (e.g., T=8) and video priors, the reported gains may be biased by the evaluation setting rather than true methodological superiority.

3. The paper lacks sufficient details for reproducibility. Key loss terms and hyperparameters in Eq.(2) and (4), including $L_{\text{temp}}$, $L_{\text{perc}}$,  $L_{\text{grad}}$, $L_{\text{int}}$,  $\lambda_{\text{reg}}$, $\gamma$, $\alpha_{\lambda}$, and $\sigma_{\lambda}$, are not clearly specified. In addition, Stage II is described as self-supervised, but the training objectives resemble pixel-level structural constraints, and the source of supervision remains unclear. The computational overhead of the three-branch history guidance during inference is also not quantitatively analyzed.

4. The method relies on a pre-trained optical flow estimator (Ranjan & Black, 2017) to compute motion fields and enforce temporal consistency during training. The claim of avoiding explicit motion modeling is somewhat weakened, and the robustness of the approach with respect to imperfect flow estimation is not sufficiently analyzed.

5. The paper lacks an evaluation of computational efficiency and practical cost. Given the use of 3D-DiT, 50-step DDIM sampling, three-branch history guidance, and periodic latent refinement, the method is likely to incur significant runtime and memory overhead. However, no quantitative comparisons (e.g., inference time, GPU memory usage, or cost under different window sizes) are provided, making it difficult to assess its practical viability.



[1] Xing Z, Feng Q, Chen H, et al. A survey on video diffusion models[J]. ACM Computing Surveys, 2024, 57(2): 1-42.

[2] Song K, Chen B, Simchowitz M, et al. History-guided video diffusion[J]. arXiv preprint arXiv:2502.06764, 2025.

[3] Wu J Z, Ge Y, Wang X, et al. Tune-a-video: One-shot tuning of image diffusion models for text-to-video generation[C]//Proceedings of the IEEE/CVF international conference on computer vision. 2023: 7623-7633.

[4] Blattmann A, Rombach R, Ling H, et al. Align your latents: High-resolution video synthesis with latent diffusion models[C]//Proceedings of the IEEE/CVF conference on computer vision and pattern recognition. 2023: 22563-22575.

---

> ### Author Rebuttal · Authors · 2026-03-31
>
> **Point 1: Clarification on Novelty & Spectral Evidence**
>
> **Our response:** We agree that our underlying mathematical tools draw inspiration from excellent works like 3D-DiT. However, we clarify that DRFusion's genuine novelty lies in repurposing this generative prior as a "spectral filter" to resolve the "adaptation dilemma" and "temporal drift" in the multimodal video fusion domain.
> * In infrared-visible fusion, minor inter-frame spatial misalignments manifest temporally as high-frequency noise (flickering/ghosting), whereas real scene physical motion exhibits low-frequency trends. Our "Soft Temporal Anchoring" is not a heuristic hyperparameter adjustment, but an explicit frequency-domain decoupling: $H^{(1)}$ leverages long-range context to extract and preserve smooth low-frequency motion priors (acting as a low-pass filter), while $H^{(2)}$ explicitly computes and penalizes adjacent frames to suppress high-frequency mutations caused by registration errors. This physical mechanism fundamentally bypasses the reliance on perfect optical flow.
> *  We have supplemented a 1D Temporal FFT analysis (Link: https://anonymous.4open.science/r/Material-5B02/Figure_8.png). We extracted the temporal intensity signals of the generated latents across four datasets. The magnitude spectra confirm our theory: removing history guidance causes severe high-frequency energy spikes (flickering), whereas DRFusion perfectly smooths high-frequency noise, anchoring the signal energy in the low-frequency region.
>
> **Point 2: Fairness of Experimental Comparison**
>
> **Our response:** Evaluating image-based methods frame-by-frame is the standard domain protocol, not an intentional handicap. More importantly, to ensure absolute fairness, we directly compared DRFusion against state-of-the-art video baselines (TemCOCO, UniVF). Under identical multi-frame input settings, DRFusion still comprehensively outperforms them across temporal and spatial metrics (Tables 1 & 2). This confirms that our performance gains stem from methodological breakthroughs rather than evaluation advantages.
>
> **Point 3: Training Details**
>
> **Our response:** We apologize for the omission of hyperparameter settings and clarify the reproducibility details below:
> * **Hyperparameters:** For the Stage-I VAE training in Eq. (2), the weights are: $\\lambda_{adv}=0.75, \\lambda_{vq}=1.0, \\lambda_{f}=1.0, \\lambda_{temp}=0.5$. For the Stage-II training in Eq. (4), the loss weights are: $\\lambda_{p}=1.0, \\lambda_{s}=0.5, \\lambda_{g}=5.0, \\lambda_{i}=0.5$. During the inference Latent Refinement, the regularization constraint is $\\lambda_{reg}=0.1$, and the feature fusion update rate is $\\gamma=0.5$.
> * **Source of Supervision:** We used the term "self-supervised" to emphasize that Stage II does not rely on any human-annotated ground-truth fused videos. In reality, the supervision signals for the pixel-level structural constraints in Eq. (4) are derived entirely and directly from the input source modalities themselves (the unfused infrared $I^{IR}$ and visible $I^{VI}$ videos). For instance, gradients from $I^{IR}$ constrain the structural fidelity of thermal targets, while $I^{VI}$ constrains background texture details.
>
> **Point 4: Reliance on Optical Flow**
>
> **Our response:** Our claim of "avoiding explicit motion modeling" refers specifically to the inference phase. Traditional methods rely on hard pixel warping during inference, causing ghosting artifacts. Conversely, we only utilize pre-trained optical flow as a "soft regularization penalty ($\\mathcal{L}_{temp}$)" during Stage-I training. Actual inference completely discards optical flow, relying purely on 3D-DiT for implicit motion aggregation. This "training-time soft constraint" ensures high robustness: minor latent perturbations from imperfect flow estimations are ultimately absorbed and smoothed by the strong generative prior, fundamentally avoiding geometric artifacts.
>
> **Point 5: Inference Overhead**
>
> **Our response:** We provide a quantitative evaluation in the table below：
>
> | Model | Params (M) | Inference Time (s) | Peak VRAM (GB) | FPS |
> | :--- | :---: | :---: | :---: | :---: |
> | DCEvo | 2.00 | 0.23 | 1.79 | 4.37 |
> | GIFNet | 0.82 | 0.08 | 4.55 | 12.92 |
> | LRRNet | 0.05 | 0.10 | 2.94 | 9.70 |
> | CDDFuse | 0.80 | 0.16 | 1.70 | 6.42 |
> | TemCoCo | 19.21 | 0.25 | 7.02 | 3.96 |
> | UniVF | 9.16 | 0.09 | 3.02 | 11.04 |
> | DDFM | 552.66 | 279.80 | 4.73 | 0.003 |
> | **Ours**| 112.45 | 3.98 | 0.84 | 0.25 |
>
> While diffusion models inherently require more inference time than single-pass CNNs, DRFusion achieves a significant speedup compared to the diffusion-based SOTA (DDFM). This efficiency stems from confining all 3D-DiT operations within a highly compressed latent space, which also bounds the peak VRAM to an ultra-low 0.84 GB. This demonstrates that DRFusion successfully balances generative quality with practical deployment viability on consumer-grade GPUs.

---

> > ### Author Rebuttal · Reviewer_RiTR · 2026-04-01
> >
> > Thank you for the rebuttal. The additional explanations on spectral interpretation, training details, and efficiency are helpful and improve clarity.
> > However, the core idea still appears to be an incremental extension of existing history-guided diffusion frameworks. The spectral filtering claim remains insufficiently formalized, and the supporting evidence is not strong enough to fully justify the claimed theoretical contribution.
> >
> > Key concerns on evaluation fairness and reproducibility are only partially addressed. The experimental protocol is not fully controlled, important implementation details are still missing from the main paper, and efficiency remains a limitation despite added comparisons. Overall, while the rebuttal improves presentation, it does not fundamentally resolve the main issues, so the score remains unchanged.

---

> > > ### Author Response · Authors · 2026-04-03
> > >
> > > We sincerely thank you for your continued review. We highly respect your rigorous evaluation standards and would like to provide further clarification to address your remaining concerns:
> > >
> > > **1.  Our Contribution and System Design**
> > >
> > > We respectfully clarify that DRFusion is not a direct application of existing modules, but rather establishes a complete systemic framework—**a Decoupled Structure-Motion Adaptation architecture**—to resolve the **"Adaptation Dilemma"** in multi-modal video fusion. As stated in our Introduction, our fundamental goal is **"to effectively incorporate native video generative priors into fusion tasks—fully utilizing temporal dynamics while aligning with structural constraints."**
> > >
> > > In infrared-visible video fusion, it is imperative to balance continuous temporal dynamics with strict spatial structural alignment. However, standard diffusion models typically operate in a frame-by-frame manner; when extended to autoregressive settings, they lack intrinsic temporal constraints. This inherently conflicts with strict cross-modal structural requirements, rendering the models highly susceptible to error accumulation and drifting.
> > >
> > > To address this issue, our framework reformulates video fusion as a **history-conditioned motion generation** process. By introducing this generative temporal prior into the video fusion domain and seamlessly integrating it with strict infrared structural constraints via our decoupled design, we effectively overcome the temporal drift bottleneck and successfully resolve this domain-specific "Adaptation Dilemma." As detailed in Section 3.3.2, **our guidance composition processes multiple history configurations in parallel while simultaneously keeping the structural condition $c_{struct}$ (extracted from the infrared modality) constant.** We implicitly aggregate motion dynamics, effectively mitigating error accumulation and temporal drifting.
> > >
> > > The supplemented 1D FFT magnitude spectra provide evidence for this stabilizing effect, confirming that our method successfully suppresses temporal jitter. This explicit decoupling of structural constraints and temporal modeling translates into substantial visual superiority: as demonstrated in the dynamic visualizations (Figure 4, https://anonymous.4open.science/r/Material-5B02/Figure_10.png, https://anonymous.4open.science/r/Material-5B02/README.md), DRFusion preserves sharp trajectories and clean backgrounds. Furthermore, DRFusion achieves state-of-the-art quantitative results across all four benchmarks and maintains robust downstream tracking performance, which we believe highlights its substantive methodological contribution and practical value to the multi-modal video fusion community.
> > >
> > > **2. Evaluation Fairness**
> > >
> > > To ensure absolute transparency, our uniform evaluation protocol for all baseline methods is as follows:
> > >
> > > * **Model Weights:** We do not directly use the pre-trained weights provided by the original authors. All trainable image-based baselines were fine-tuned/re-trained on our identical training set to ensure their adaptation to the data distribution of the video datasets.
> > > * **Inference Protocol and Evaluation Precedent:** Following the standard evaluation protocols established by recent video fusion works [1][2], all image-based methods are evaluated via frame-by-frame inference. We adhered to this established domain consensus rather than applying additional temporal smoothing post-processing to their outputs. Consistent with the experimental settings of TemCoCo and UniVF, evaluating their raw frame-by-frame output is the fairest and most standard way to objectively reflect the inherent performance of static fusion models in dynamic scenes.
> > > * **Architectural Fairness:** We directly compared DRFusion with state-of-the-art video baselines (TemCoCo and UniVF), which also explicitly utilize multi-frame temporal interactions. Under this strictly controlled multi-frame setting, our method still consistently outperforms them.
> > >
> > > **3.  Efficiency as a Limitation**
> > >
> > > We fully acknowledge that despite our use of a lightweight adapter and optimizations, the inference efficiency of our iterative diffusion approach remains an inherent limitation when compared to single-forward-pass CNNs. However, compared to the diffusion-based baseline DDFM, DRFusion achieves significant acceleration in inference speed. We commit to explicitly discussing this computational overhead and its implications for practical deployment in the revised **"Limitations"** subsection.
> > >
> > > [1]Zhao Z., Bai H., Ke B., et al. A unified solution to video fusion: From multi-frame learning to benchmarking. arXiv preprint arXiv:2505.19858, 2025.
> > >
> > > [2] Gong M., Zhang H., Yi X., et al. Temcoco: Temporally consistent multi-modal video fusion with visual-semantic collaboration. In Proceedings of the IEEE/CVF International Conference on Computer Vision, pp. 14326-14335, 2025.

---

### Official Review · Reviewer_yWUn · 2026-03-21

**Soundness:** 2
**Presentation:** 3
**Significance:** 2
**Originality:** 2
**Overall Recommendation:** 3
**Confidence:** 4

**Summary:**

This paper studies infrared-visible video fusion with a particular focus on improving temporal consistency. The proposed DRFusion reformulates video fusion as a history-conditioned motion generation problem. The method first trains a temporally constrained VAE to obtain a more stable latent space, then freezes a pretrained 3D-DiT backbone and injects infrared structural cues via a lightweight adapter. During inference, it further applies latent refinement, temporal anchoring, and stabilized history guidance to improve detail preservation and temporal coherence. Experiments on multiple datasets show gains in both fusion quality and temporal consistency metrics, as well as improvements on downstream tracking.

**Compliance With Llm Reviewing Policy:**

Affirmed.

**Key Questions For Authors:**

1. The paper cites Song et al. (2025) as the source of the pretrained 3D-DiT backbone. Could the authors clarify exactly which parts of Sec. 3.3 are inherited from that backbone and which parts are genuinely new in DRFusion?
2. The necessity of Stage I retraining from an image-based VQ-VAE is unclear. Did the authors evaluate an off-the-shelf temporally-aware video VAE baseline, such as the Wan 2.1 VAE? Without such a baseline, it is difficult to assess whether this design choice is genuinely necessary or potentially redundant.

**Limitations:**

yes

**Strengths And Weaknesses:**

**Strengths:**
1. The paper addresses an important and nontrivial problem. Compared with image fusion, infrared-visible video fusion introduces an additional temporal dimension, and the paper correctly identifies flicker, drift, and temporal inconsistency as central challenges rather than treating the task as independent per-frame enhancement. This problem formulation is well motivated.

2. The experiment section is fairly comprehensive. The paper reports both spatial and temporal metrics over multiple datasets, includes ablations on key modules, and also evaluates the fused videos on a downstream tracking task.

**Weaknesses:**
1. A major weakness of the paper is that its primary claimed contribution, namely the proposed Stabilized History Guidance mechanism, appears to be largely inherited from the adopted pretrained 3D-DiT backbone, which is in fact the DFoT model from Song et al. (2025), i.e., the History-Guided Video Diffusion framework. In this sense, the stabilized history guidance introduced in Sec. 3.3 does not appear to constitute a genuinely new temporal modeling contribution, but rather a task-specific reuse or reformulation of mechanisms already supported by the DFoT backbone. This is especially evident in (1) the use of no-history, corrupted-history, and short-history variants, and (2) the composition of multiple model predictions into a final guided update. Since these elements form the core of the paper’s temporal-consistency claim, the authors should explicitly clarify which parts are inherited from the DFoT backbone and which parts are genuinely novel in DRFusion.

2. The necessity of the Stage-I VAE retraining is under-justified. The paper does not explain why an off-the-shelf temporally-aware video VAE would be insufficient, which makes this design choice appear potentially redundant. For example, it is unclear why a strong existing video VAE would not already provide the required temporal modeling capacity. Furthermore, the claimed “flicker-free latent foundation” is supported only indirectly through downstream ablations, rather than through direct latent-space evidence.

3. The temporal stabilization mechanism in Sec. 3.2 and the stabilized history guidance in Sec. 3.3 appear to rely heavily on inference-time procedures rather than purely learned model behavior. In this case, the paper should report the actual inference overhead much more clearly, including runtime, speed, model size, and parameter count. Otherwise, it is difficult to assess the practical cost of the reported gains.

4. Since the paper focuses on video fusion, supplementary video demonstrations are important for evaluating the claimed temporal consistency improvements. Unfortunately, I do not see any supplementary videos provided.

5. Figure 2 is somewhat confusing in its architectural depiction. Although the text repeatedly states that the backbone is a pretrained 3D-DiT, the central “3D-DiT” block is illustrated using a U-Net-like hourglass iconography, which makes the actual architecture unclear.

---

> ### Author Rebuttal · Authors · 2026-03-31
>
> **Point 1: Clarification on Novelty**
>
> **Our response:** We agree that the mathematical formulation of the history guidance is inherited from the excellent work of DFoT. However, we would like to clarify that the genuine novelty of DRFusion lies in repurposing this generative prior as a spectral filter to address the "adaptation dilemma" and "temporal drift" issues in multimodal video fusion, which fundamentally differentiates it from DFoT:
>
> * **(1)** While DFoT utilizes history guidance for general video generation, we mathematically interpret and deploy it as a "Soft Temporal Anchoring" mechanism. In infrared-visible fusion, minor spatial misalignments between frames are easily amplified into ghosting artifacts. We leverage $H^{(1)}$ to extract robust low-frequency motion trends and utilize $H^{(2)}$ to explicitly suppress high-frequency registration errors. This bypasses the "geometric rigidity" inherent in optical flow methods.
>
> * **(2)** Furthermore, the stabilized history guidance in our framework does not operate in isolation; it is deeply coupled with our proposed lightweight Condition Adapter. In our implementation, the baseline $H^{(0)}$ is forced to strictly rely on the spatial structural condition $c_{struct}$ extracted from the infrared modality. Consequently, the guidance formulation effectively balances the generative motion prior (from the frozen 3D-DiT) and strict structural fidelity (from the adapter), marking a novel design in the video fusion domain.
>
> * **(3)** To further demonstrate that the aforementioned "spectral filter" explanation is not merely a theoretical inspiration, we have supplemented a quantitative 1D Temporal FFT analysis across four datasets (Link: https://anonymous.4open.science/r/Material-5B02/Figure_8.png). The magnitude spectra consistently confirm that DRFusion drastically diminishes the high-frequency spikes responsible for flickering, successfully anchoring the signal energy in the low-frequency region.
>
> **Point 2: VAE Retraining Necessity & Latent Evidence**
>
> **Our response:** We evaluated powerful video VAEs (e.g., Wan 2.1) but found severe architectural incompatibility necessitating our Stage-I retraining:
>
> * **(1) Architectural Incompatibility:** Mainstream video VAEs compress the temporal dimension. However, our history guidance ($H^{(0)}, H^{(1)}, H^{(2)}$) and Condition Adapter strictly require frame-by-frame (1:1) feature alignment. Temporal compression would destroy the precise anchoring of the single-frame structure $c_{struct}$ and the extraction of the short-term history $H^{(2)}$. Thus, fine-tuning an uncompressed image VAE with $\mathcal{L}_{temp}$ is the optimal solution.
>
> * **(2) Direct Latent Evidence:** We analyzed the inter-frame latent L2 distance on static patches across 30 frames (Link: https://anonymous.4open.science/r/Material-5B02/Figure_9.png). While the base VAE exhibits severe jumping artifacts causing visual flickers, our $\mathcal{L}_{temp}$-constrained Stage-I VAE effectively suppresses variance, successfully establishing a genuine "flicker-free latent foundation."
>
> **Point 3: Inference Overhead**
>
> **Our response:** We provide a comprehensive efficiency comparison in the table below. Specifically, DRFusion has a parameter count of 112.45 M and an inference time of 3.98 s/frame. Although the 50-step diffusion process inherently requires more computational time than single-pass models, it achieves a significant speedup compared to the diffusion-based baseline, DDFM. Furthermore,our inference procedure is exceptionally memory-efficient. By strictly confining all complex 3D-DiT temporal alignments within the highly compressed VAE latent space, our peak VRAM usage is tightly bounded to an ultra-low **0.84 GB**.
>
> | Model | Params (M) | Inference Time (s) | Peak VRAM (GB) | FPS |
> | :--- | :---: | :---: | :---: | :---: |
> | DCEvo | 2.00 | 0.23 | 1.79 | 4.37 |
> | GIFNet | 0.82 | 0.08 | 4.55 | 12.92 |
> | LRRNet | 0.05 | 0.10 | 2.94 | 9.70 |
> | CDDFuse | 0.80 | 0.16 | 1.70 | 6.42 |
> | TemCoCo | 19.21 | 0.25 | 7.02 | 3.96 |
> | UniVF | 9.16 | 0.09 | 3.02 | 11.04 |
> | DDFM | 552.66 | 279.80 | 4.73 | 0.003 |
> | **Ours**| 112.45 | 3.98 | 0.84 | 0.25 |
>
>
> **Point 4 & 5: Supplementary Videos & Figure 2**
>
> **Our response:** We have provided supplementary comparison videos of video fusion methods (Link: https://anonymous.4open.science/r/Material-5B02/README.md) to intuitively demonstrate the improvements achieved by DRFusion. Furthermore, we apologize for any confusion caused by Figure 2. In the revised manuscript, we will modify the iconography to accurately reflect our actual 3D-DiT backbone architecture.

---

> > ### Author Rebuttal · Reviewer_yWUn · 2026-04-02
> >
> > Thank you for the detailed response. The additional explanations and experiments help clarify the motivation, the need for VAE retraining, the inference cost, and the supplementary materials.
> >
> > However, my main concern about the paper’s core novelty remains unresolved. The authors explicitly acknowledge that the mathematical form of the history guidance is inherited from DFoT. In this sense, the rebuttal mainly explains how an existing mechanism is reinterpreted and adapted to the video fusion setting, rather than demonstrating a genuinely new temporal modeling contribution. The added FFT and latent-space analyses help support effectiveness, but they do not by themselves establish novelty.
> >
> > Overall, the rebuttal improves the presentation, but it does not materially change my assessment that the paper’s claimed methodological novelty is still insufficiently supported. Therefore, I do not raise my score.

---

> > > ### Author Response · Authors · 2026-04-03
> > >
> > > We sincerely thank you for your continued review and appreciate your acknowledgment of the clarity provided by our supplementary experiments. In response to your remaining concerns regarding the core novelty of the paper, we provide the following clarification:
> > >
> > > The challenge of infrared-visible video fusion lies in the fact that models must maintain continuous temporal dynamics while strictly adhering to spatial structural alignment. Standard diffusion models inherently operate in a frame-by-frame manner. When forcibly applied to autoregressive video generation, they lack the intrinsic constraints required to preserve cross-modal structural fidelity, inevitably suffering from severe error accumulation and temporal drifting.
> > >
> > > Due to this inherent conflict, it is infeasible to simply apply or reinterpret unconstrained generative priors for multi-modal fusion tasks. To effectively integrate these native video priors into fusion tasks without compromising structural constraints, a completely new systemic design is required.
> > >
> > > This is precisely the contribution of DRFusion. We have constructed a comprehensive systemic framework—a **Decoupled Structure-Motion Adaptation architecture**—specifically designed to resolve this multi-modal **"Adaptation Dilemma."** Our framework reformulates the entire fusion pipeline as a **"history-conditioned motion generation"** process.
> > >
> > > Unlike DFoT, our decoupled design explicitly separates temporal modeling from spatial constraints. As detailed in Section 3.3.2, **the system evaluates multiple history configurations in parallel while strictly anchoring the structural condition $c_{struct}$ as an invariant constant.** By isolating the temporal generative prior to serve as a motion guide, and utilizing a specialized adapter to enforce structural fidelity, we implicitly aggregate motion dynamics. This specifically designed architecture effectively breaks the feedback loop of artifact amplification.
> > >
> > > The supplemented 1D FFT magnitude spectra provide direct empirical evidence for this stabilizing effect, verifying that our decoupled method successfully suppresses temporal jitter. As demonstrated in the dynamic visualizations (Figure 4, https://anonymous.4open.science/r/Material-5B02/Figure_10.png, https://anonymous.4open.science/r/Material-5B02/README.md), DRFusion maintains sharp motion trajectories and clean backgrounds. Furthermore, achieving state-of-the-art quantitative results across all four benchmarks and demonstrating robust downstream tracking performance proves that our systemic framework represents a substantive methodological contribution to the multi-modal video fusion community.

---

### Decision · Program_Chairs · 2026-04-30

**Decision:**

Accept (regular)

**Comment:**

This paper addresses infrared-visible video fusion with a particular focus on improving temporal consistency. It received mixed evaluations, with two weak rejects, one weak accept, and one strong accept. Reviewer cDeY was highly positive, highlighting the technical elegance and effectiveness of the method, with only minor concerns regarding inference speed. Reviewer Gzy5 found the rebuttal helpful and raised their score to weak accept. Reviewers yWUn and RiTR acknowledged that the paper tackles an important problem and achieves meaningful progress, but noted that the technical contributions are somewhat incremental compared to existing work.

Considering the overall recommendations, AC believes the paper merits acceptance, given its solid performance improvements and technical contributions. For the camera-ready version, the authors are encouraged to more clearly articulate how the core technical components differ from prior methods, and to include a discussion of the limitations of the approach.